# Spatial Mixture-of-Experts

**Nikoli Dryden**
ETH Zürich
ndryden@ethz.ch

**Torsten Hoefler**
ETH Zürich
htor@inf.ethz.ch

## Abstract

Many data have an underlying dependence on spatial location; it may be weather on the Earth, a simulation on a mesh, or a registered image. Yet this feature is rarely taken advantage of, and violates common assumptions made by many neural network layers, such as translation equivariance. Further, many works that do incorporate locality fail to capture fine-grained structure. To address this, we introduce the Spatial Mixture-of-Experts (SMoE) layer, a sparsely-gated layer that learns spatial structure in the input domain and routes experts at a fine-grained level to utilize it. We also develop new techniques to train SMoEs, including a self-supervised routing loss and damping expert errors. Finally, we show strong results for SMoEs on numerous tasks, and set new state-of-the-art results for medium-range weather prediction and post-processing ensemble weather forecasts.

## 1 Introduction

Many datasets exhibit an underlying, location-based structure, where the value at a point depends on where that point is. For example, weather predictions [78] depend on their location on Earth; many scientific simulations are parameterized on an underlying mesh [9]; data (e.g., faces) may be specifically aligned [91]; or it may approximately hold, as in natural images with centered objects. Further, tasks on such data are often dense regression tasks, such as predicting weather several days in the future. Numerous architectures have been successfully applied to such tasks. Convolutional neural networks (CNNs) [80] and transformers [74] show promise for medium-range weather prediction. Locally-connected networks (LCNs), which use independent, rather than shared, filters at each point, have been applied to weather post-processing [38], face recognition [46, 77, 91], and other tasks [18, 71], specifically to learn local features. Low-rank local connectivity (LRLCN) [30] relaxes the translation equivariance of CNNs while requiring fewer parameters than LCNs. Other approaches, such as CoordConv [65], add an additional inductive bias by providing explicit input coordinates.

However, for tasks on data with location-based structure, prior approaches suffer from various limitations. Convolution assumes that data is translation equivariant, which does not hold for such tasks [46, 91]. Approaches like LCNs require many parameters. Many architectures have been designed for classification tasks and fail to perform well on regression because they opererate at too coarse granularity and are unable to capture key details. This limits their applicability to important tasks, such as medium-range weather prediction or climate simulations. Indeed, on a simple heat diffusion task with location-dependent diffusivities (see §3.1), many approaches do not learn the location dependence at all and instead converge to an "average" diffusivity.

To address this, we introduce a novel neural network layer, the Spatial Mixture-of-Experts (SMoE) layer (§2). An SMoE uses a learned gating function to sparsely select and route from a shared set of experts to each spatial location (e.g., pixel) in an input sample. This enables experts to specialize to the unique characteristics of different "regions" within the data and easy interpretability by analyzing the gate. SMoEs require the assumption that all samples in the dataset have a similar underlying spatial structure, but this often holds (at least approximately) for many datasets, such as weather, where each example is on the same latitude/longitude grid. For the SMoE gating function, we introduce *tensor routing*, a simple and cheap routing function that effectively learns this spatial

36th Conference on Neural Information Processing Systems (NeurIPS 2022).

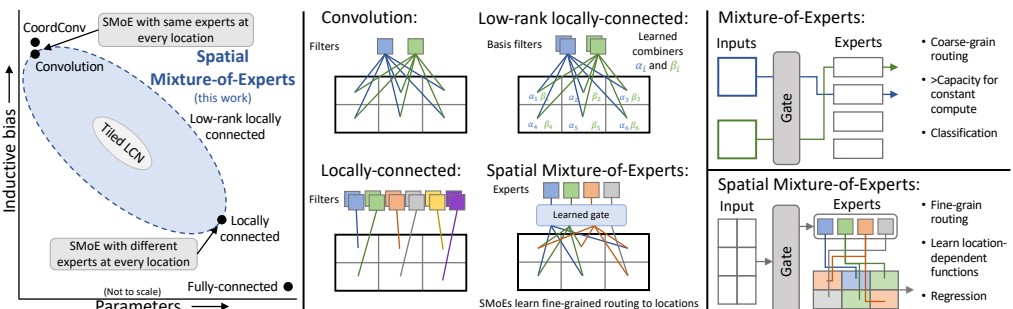

Figure 1: Comparison of SMoEs to other networks. **Left**: Qualitative comparison of parameter counts and inductive biases. **Center**: Pattern of filter application among models. **Right**: Routing in MoEs and SMoEs.

structure (§2.1). We also find that end-to-end training approaches, as in standard mixture-of-experts (MoEs) [82, 86], do not capture location-dependence well. Instead, we introduce a separate *routing classification loss* (§2.2) to train the gate, a self-supervised loss which provides a better learning signal to the gate about incorrect routings. We also introduce *expert error damping* (§2.3), which limits expert updates from misroutings, and further improves performance. Both methods rely on extracting information from the error signal (gradient of the layer output) which would otherwise be uninformative. Figure 1 provides an overview of SMoEs and a qualitative comparison to important related work. With these methods, SMoEs are able to learn fine-grain, location-dependent structure which other architectures miss, and consequentially deliver much better performance.

SMoEs are related to and inspired by prior work in MoEs (e.g., [82, 86, 102]), but there are key differences. Existing MoEs use coarse-grained routing at the sample [99], token [86], or patch [82] level, and experts produce coarse output (e.g., entire samples or channels) while SMoEs route at a fine-grained, per-pixel level. Fine-grained routing is important for enabling experts to specialize to fine-scale information, which may be crucial [74, 84]. Such MoEs also typically aim to increase model capacity while keeping compute costs constant, whereas the goal of SMoEs is to capture spatial dependencies. Other work has aimed to incorporate spatial dependence, such as LRLCNs [30], which learn content- and location-dependent weights to combine a set of basis filters. However, we found that these prior methods failed to capture the fine-grained features necessary for good performance on dense regression tasks (§3.1).

We conduct experiments on several benchmark datasets with SMoEs and conduct extensive ablation studies of SMoE design decisions (§3). Using a simple heat diffusion dataset, we showcase the limitations of other models and the power of SMoEs in a controlled situation. We then apply SMoEs to medium-range weather prediction and outperform the state-of-the-art [53, 80] on WeatherBench [78]; and set a new state-of-the-art for post-processing ensemble weather forecasts on the ENS-10 dataset [8]. Finally, we show that SMoEs can also be applied to image classification tasks, where we match or outperform LRLCNs while using fewer parameters.

Our code is available at https://github.com/spcl/smoe.

## 1.1 Related Work

**Mixture-of-Experts and conditional computation.** While MoEs have long been used [15, 49, 52], since the advent of deep learning, there has been much work in applying MoEs, conditional computation, and dynamic routing to DNNs [1, 4, 5, 7, 10–12, 19–21, 23, 25, 27, 29, 33, 34, 39, 51, 59, 60, 70, 75, 76, 82, 83, 86, 99, 102, 107]. Often, the goal is to increase model capacity without a corresponding increase in compute costs and models use coarse-grained routing. Notably, MoEs route at the sample (e.g., [25, 29, 70, 83]), token (e.g., [33, 59, 86]), or patch (e.g., [82]) level; are typically trained with extensive auxiliary losses; and often target classification tasks. This coarse routing, in particular, means there is limited opportunity for experts to specialize. MoEs specifically for vision tasks also typically operate at a sample-level (e.g., [1, 4, 39, 99, 102]) and use experts to specialize filter or channel choices. In contrast, our SMoEs induce an inductive bias via fine-grained, location-dependent routing for problems with such underlying structure; typically do not need auxiliary losses beyond the routing classification loss; and work well for dense regression tasks.

**Local spatial structure.** Locally connected networks (i.e., convolution filters without weight sharing) were historically successful in vision tasks such as facial recognition [18, 46, 77, 91]. Further

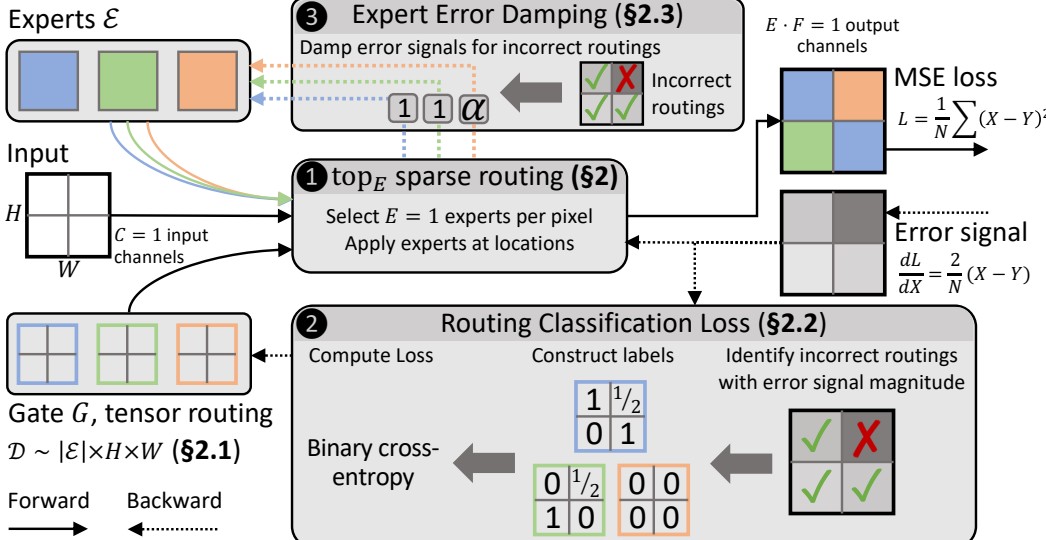

Figure 2: Overview of SMoE architecture and training using a mean-square error loss and an example input. ❶ Each location in an input is routed to $E$ experts from a set $\mathcal{E}$ based on a gate $G$ (§§2 and 2.1). ❷ The gate is trained using a routing classification loss, which identifies incorrect routings based on the error signal (§2.2). ❸ The error signal to experts is damped for locations they were incorrectly routed to (§2.3).

work incorporated periodic or tiled weight sharing [37, 71, 105]. However, more recent work often exclusively uses convolution, which has been found to perform better [72] while requiring significantly fewer parameters. CoordConv [65] explicitly provides pixel coordinates as input. Because of their structure, a single convolution or CoordConv layer cannot learn location-dependent sparse activations as SMoEs do, and multi-layer networks give limited improvements in practice. Low-rank locally connected networks [30] learn to combine a small number of basis filters in a location- and input-dependent manner. These combiners lack sparsity and apply all basis filters at each point, and are softmax-normalized, which can result in a few filters dominating, limiting diversity. Separate work has used attention to provide position-dependent scaling or integrate context [35, 50, 64, 93, 95, 100]. This has culminated in vision transformers [26] and variants [28, 62, 67, 96, 97], which use specialized attention mechanisms to integrate spatial information. Other work uses Markov [61, 66] or conditional random fields [16, 43, 106], or graph-based methods [17, 98, 104], to learn long-range dependencies. Additional work has studied incorporating equivariance properties into neural networks (e.g., [13]).

**Sparsity.** Our use of sparse routing resembles work on sparsity in DNNs in general [45]. Many approaches learn a fine-grained mask to identify which weights to keep or prune (e.g., [63, 90]), while other approaches sparsify activations (e.g., [3, 47, 56, 68, 69]). These works use sparsity to improve runtime performance, typically at inference time.

**Weather prediction.** Medium-range weather prediction, or forecasting roughly three to seven days in advance [6], is of broad societal import [57]. There has been much interest in applying deep learning to this task [85], and WeatherBench [78] serves as a community benchmark. Deep learning has also been successfully applied to the related tasks of ensemble post-processing [38, 79] and now-casting (forecasting a few hours in advance) [2, 31, 81, 87–89]. Standard CNNs are currently state-of-the-art on WeatherBench [80] (although graph neural networks show promise on similar data [53]) and are competitive on post-processing tasks [8]. SMoEs can take advantage of the extensive fine-grained, location dependent structure in weather data for improved performance.

## 2 Spatial Mixture of Experts

We now introduce the Spatial Mixture-of-Experts (SMoE) layer. An SMoE uses a learned gating function (§2.1) to select specific experts from a shared set to apply at each point (e.g., pixel) in an input. The gate learns the underlying spatial structure of the data, and routes specific experts to different "regions", allowing them to specialize. An SMoE is predicated on the assumption that the spatial structure is similar across all samples in a dataset. We also assume data are on a Cartesian mesh (i.e., grid), although this could be generalized. To train SMoEs, we introduce a self-supervised

*routing classification loss* (§2.2) and *expert error damping* (§2.3), which we found essential for achieving the best performance. Figure 2 provides an overview of SMoEs and their training.

We first define the SMoE layer in full generality, then discuss the particular implementation we use. (See §3.1 for ablations.) Let $x \in \mathbb{R}^{C \times H \times W}$ be an input sample (we assume 2D data for simplicity). The SMoE layer consists of a set $\mathcal{E}$ of experts, of which $E \leq |\mathcal{E}|$ will be selected at each point, and a gating function $G : \mathbb{R}^{C \times H \times W} \to \mathbb{R}^{|\mathcal{E}| \times H \times W}$. The gating function has exactly $E$ nonzeros at each spatial point ($H \times W$), which correspond to the selected experts at that point. Each expert $e \in \mathcal{E}$ is applied at the points in $x$ where it has been selected, and may include additional context from $x$ (e.g., surrounding points if $e$ is convolution), and produces an output of size $F$. The outputs of each expert at a point are then concatenated to form the output channel dimension (of size $E \cdot F$) of the SMoE. More precisely, let $\mathrm{gather}_I(\cdot)$ select only the input entries where $I$ is nonzero. Then an SMoE is:

$$y = \mathrm{gather}_{G(x)}\big(G(x) \cdot [e_1(x); \ldots; e_{|\mathcal{E}|}(x)]\big),$$

where $\cdot$ is element-wise multiplication, $[e; \ldots]$ stacks tensors, and $y \in \mathbb{R}^{EF \times H \times W}$. When $E \ll |\mathcal{E}|$, the gating function induces significant sparsity; this allows an SMoE to compute experts only at the points where they are to be applied, avoiding most computation.

This formulation yields a *weighted* SMoE, where the expert outputs are scaled by the gating function. An alternative, which is slightly more efficient and can be more readily interpretable, is an *unweighted* SMoE, in which case experts are not scaled, and the $\mathrm{gather}$ only uses $G(x)$ to select experts.

In this work, we focus on a simple, yet powerful, SMoE layer using convolution filters as experts:

$$y_i = G(x)_i \cdot \sum_{c \in [C]} w_{i,c} \star x_c,$$

where $\star$ is cross-correlation, $i \in [|\mathcal{E}|]$, and we have elided the gather for simplicity.

## 2.1 Gating Functions

The gating function $G$ in an SMoE is critical for learning the underlying location-dependence of the data. While $G$ can be any function, a good gate should be relatively cheap. We follow prior work on MoEs (e.g., [82, 86]) and use top-$E$ routing to select experts and sparsify the gate: $G(x) = \mathrm{top}_E(g(x))$, where $g(x) \in \mathbb{R}^{|\mathcal{E}| \times H \times W}$ is a learnable gating layer and $\mathrm{top}_E$ selects the $E$ largest entries along the first (expert) dimension and sets the remaining entries to 0. However, we found that normalizing the gating function using softmax, as is typically done, was not necessary.

There are many options for the gating layer. Many MoE works have used MLPs [82, 86], but we found this did not perform well. Convolution or CoordConv [65] also did not perform well (§3.1). Instead, we find that a simple *tensor routing* gating layer worked best. With this, $g(x) = \mathcal{D} \in \mathbb{R}^{|\mathcal{E}| \times H \times W}$, where $\mathcal{D}$ is a learnable tensor that directly encodes the underlying location dependence and routing structure without depending on the input. Further, $\mathcal{D}$ uses one parameter per expert per location and requires no computation beyond optimizer updates, making it an efficient choice.

We initialize $\mathcal{D}$ using a uniform distribution over $[-3|\mathcal{E}|/EF, 3|\mathcal{E}|/EF]$, which corresponds to a Kaiming uniform initialization with fan-in $EF/|\mathcal{E}|$ [41]. However, in many practical cases, some data about locations may be known (e.g., whether it is land or sea when doing weather prediction). In such cases, $\mathcal{D}$ can be initialized based on this data to assign groups of experts in advance. This allows the gating function to benefit from prior knowledge while still being able to adjust the routing. Additionally, a network may contain many SMoE layers, in which case we can share $\mathcal{D}$ between layers that have the same spatial dimensions and number of experts, which reduces the overhead of the gate.

Finally, many MoE formulations use a number of auxiliary losses to ensure gates learn good routing policies and avoid mode collapse (e.g., [10, 82, 86]). When training with the routing classification loss we propose, we did not find additional auxiliary losses to be necessary.

## 2.2 Training and the Routing Classification Loss

Training MoEs is typically done end-to-end, with the experts and gate learning simultaneously, and sparse gradients based on the routing. We found this did not lead to good performance with SMoEs and tensor routing, particularly on regression tasks, as the gate did not learn well: it rarely changed its routing decisions from initialization. We hypothesize that this is due to a "mismatch" in

the gradients on regression tasks, where they are informative for experts but not the gate, because regression aims to make a continuous prediction over both positive and negative values, whereas selecting an expert requires a threshold (see §B). To address this, we train the gate with a separate, self-supervised loss function, the *routing classification* (RC) loss. The key idea is to consider routing as a dense, multi-label classification task: selecting the "correct" experts at each point (cf. semantic segmentation). The RC loss does exactly this, and trains the gate by constructing appropriate labels. This also helps avoid mode collapse, where only a small number of experts are selected (see §3.2).

In order to construct these labels, we need to determine whether the gate selected the correct set of experts at each point. However, such information is not directly available. Instead, we use the error signal into the SMoE layer (i.e., the gradient of the layer output w.r.t. the loss) as a proxy, and say that the routing was incorrect when the error signal has a large magnitude for the expert at that point, as this will imply a correspondingly large gradient update. Further, in the case of a mean-square error loss $L$ (commonly used for regression), the error signal can directly encode the prediction error. Let $X$ be the predictions, $Y$ the true values, and $N$ the number of elements in $X$. Then the error signal is:

$$\frac{dL}{dX} = \frac{d}{dX} \frac{1}{N} \sum (X - Y)^2 = \frac{2}{N}(X - Y).$$

Hence, the error signal is simply the (scaled, signed) error of the predictions. While this exact relation ceases to hold as backpropagation proceeds, the intuition behind the error signal magnitude remains.

We now define the routing classification loss. Given the error signal $\varepsilon$ into an SMoE layer, and an error quantile $q$ (a hyperparameter), we say that selecting an expert at a point was incorrect if $\varepsilon$ at that expert and point is greater than the $q$th quantile of $\varepsilon$, and correct otherwise. We use quantiles as they are independent of $\varepsilon$'s scale, which may be hard to determine and change during training. We then construct the labels for each point as follows: Unselected experts start with label $0$. A correctly selected expert has label $1$. Finally, if an expert was incorrectly selected, its label is $0$ and we add $1/(|\mathcal{E}| - E)$ to the label value of each unselected expert (note $|\mathcal{E}| - E$ is the number of unselected experts). This corresponds to a uniform prior that the correct expert could be any unselected expert. With these labels, the RC loss for a gate is then the binary cross-entropy loss of the gate output.

## 2.3 Expert Error Damping

While the RC loss enables an SMoE gate to be trained directly based on whether it routed correctly, experts that were incorrectly routed to still perform gradient updates based on this routing. This results in experts updating to improve their performance for locations they may not be applied at in future training iterations after routing changes. To mitigate this, we propose *expert error damping*, where the portion of an error signal that corresponds to incorrect routings is damped to limit incorrect updates. We find that this can improve SMoE performance and reduce convergence time.

Expert error damping is similar to the RC loss, and we classify incorrect routings in the same way. We then scale the error signal into the experts by a constant factor at each point where the routing was incorrect. This will limit the magnitude of the update made in response to the misrouting.

## 2.4 Practical Implementation

We now discuss the implementation of an SMoE layer, primarily focusing on the simple SMoE with convolution filters we use. In an ideal implementation, the flop cost of an SMoE is the cost of the gate plus the cost of applying the selected experts. When using a tensor routing gate, there are no flops in the gate. The flops from applying the selected convolutional experts is equivalent to a standard convolutional layer with a number of filters equal to the number of selected experts. Hence, SMoEs are quite efficient flop-wise. However, recent work has shown that data movement is also critical to performance [48]. Because of this, we do not expect even well-optimized implementations to match the runtime of a convolution layer due to the sparse routing, limited locality in accessing expert filters, and other operations, although work on hardware-accelerated sparsity [22] should offer benefits. Despite this, during inference additional optimizations may be available because the tensor routing gate does not depend on the input and computations could be reordered to maximize locality.

Unfortunately, efficiently implementing the irregular access patterns in an SMoE is challenging in standard Python frameworks, and likely requires custom compute kernels. Instead we opt for a naïve implementation in PyTorch [73] where we apply all experts at all points and then use `gather` and `scatter` operations to implement sparse routing. Experts are concatenated in sorted routing score

| Model | #Params | Epochs | % within 1% |
|---|---|---|---|
| Convolution | 146 | 102 | $91.3^{\pm0.2}$ |
| CoordConv [65] | 56 | 50 | $91.1^{\pm0.2}$ |
| CondConv [102] | 200 | 120 | $91.2^{\pm0.3}$ |
| LRLCN [30] | 516 | 27 | $91.8^{\pm0.5}$ |
| LRLCN-ND [30] | 12315 | 35 | $91.5^{\pm0.3}$ |
| LCN | 36764 | 110 | $\mathbf{100.0}^{\pm0}$ |
| FC | 16.7M | 250 | $14.2^{\pm1.2}$ |
| ViT [26] | 200k | 67 | $93.5^{\pm0.1}$ |
| V-MoE [82] | 470k | 74 | $93.8^{\pm0.3}$ |
| SMoE | 27+12288 | 8 | $\mathbf{100.0}^{\pm0}$ |

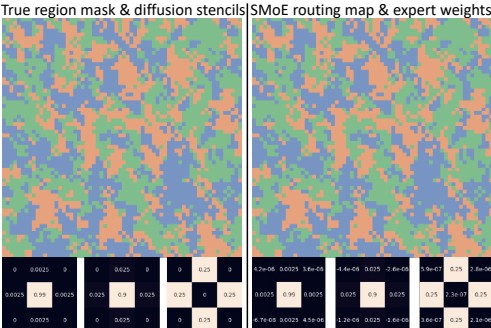

Figure 3: Heat diffusion results. **Left**: SMoE and baseline performance. **Center**: Dataset region map and diffusion stencils. **Right**: SMoE routing map and experts. SMoEs learn correct stencils and location-dependence.

order (i.e., as given by $G(x)$); while this can result in channel orders changing during training, we find it has little impact (see §3.2). We find this implementation sufficient even for large-scale tasks.

# 3 Experiments

We now present experimental results using SMoEs and a variety of baseline models, as well as ablations of different SMoE components. First, we describe a simple location-dependent heat diffusion dataset, which we use to study the ability of different architectures to learn location dependence in a controlled environment (§3.1). We then present results on the WeatherBench [78] medium-range weather forecasting challenge (§3.2) and the ENS-10 [8] dataset for post-processing ensemble weather forecasts (§3.3), where SMoEs set new state-of-the-art performance results. Lastly, we show results on several image classification tasks, illustrating the breadth of SMoE applicability even in situations without strict location-dependence (§3.4).

All results were run using PyTorch [73] version 1.11 on a large cluster with 16 GB V100 GPUs. We summarize training details throughout this section, and provide full details in §A. We use existing PyTorch implementations when available, and implemented methods ourselves otherwise. Unless noted, all SMoE results used 3×3 convolution kernel experts, unweighted tensor routing gates, RC loss, expert error damping, error quantile $q = 0.7$, and damping factor 0.1. We report the mean and standard deviation (mean$^{\pm\text{std}}$) over ten runs for all experiments, except for WeatherBench and ImageNet, which used three runs. In total, we used about 30k GPU hours for experiments.

## 3.1 Location-Dependent Heat Diffusion

To study location-dependence in a controlled manner, we generated a heat diffusion dataset with location-dependent diffusivities. The task is to predict what the heat will be at the next timestep, given the current heat at each point. Because we know the exact diffusivities and their distribution in the data, it is easy to identify how well a model has learned this task. We first describe the dataset and its generation in detail, then discuss results and ablation studies on SMoE components.

**Dataset.** The dataset consists of a region map, where each location is assigned a type, and each type corresponds to a different heat diffusivity. The region map and diffusivity assignment are then fixed for the dataset. We generate the region map randomly using a simple preferential attachment method. This defines the location-dependence that we wish to learn. To generate samples, we first randomly distribute drops of heat across the domain, then apply five-point diffusion stencils at each point, using the diffusivity of the region type for each point. For simplicity, we use zero boundary conditions. This process is iterated to generate many timesteps from the given starting state. The dataset then consists of the generated timesteps for many different starting states.

If a model is able to learn the diffusion stencils and the location-diffusivity correspondence, it can exactly predict the next timestep. Further, the diffusivity stencils are also simple, and exactly correspond to a 3×3 convolution kernel with no nonlinearity.

The particular dataset we use consists of 100,000 64×64 samples, with 1,000 initial states evolved for 100 timesteps each. Adding more samples did not significantly change results. There are three region types, with diffusivities 0.25, 0.025, and 0.0025. Figure 3 (center) shows the region map and

Table 1: Effect of RC loss and expert error damping on the heat diffusion dataset.

| RC loss | Damping | Epochs | % within 1% |
|---------|---------|--------|-------------|
| ✗ | ✗ | 14 | $91.5^{\pm 0.4}$ |
| ✗ | ✓ | 16 | $91.6^{\pm 0.2}$ |
| ✓ | ✗ | 15 | $96.7^{\pm 0.4}$ |
| ✓ | ✓ | 8 | $\mathbf{100.0}^{\pm 0}$ |

Table 2: Effect of different SMoE gate functions on the heat diffusion dataset.

| Gate | #Params | % within 1% |
|------|---------|-------------|
| Fully-connected | 50M | $85.2^{\pm 5.7}$ |
| 3×3 convolution | 27 | $91.3^{\pm 0.3}$ |
| 3×3 CoordConv [65] | 81 | $91.5^{\pm 0.4}$ |
| 3×3 LCN | 110592 | $96.3^{\pm 1.2}$ |
| 3×3 CoordConv×3 | 255 | $91.6^{\pm 0.2}$ |
| Tensor routing | 12288 | $\mathbf{100.0}^{\pm 0}$ |

diffusion stencils. We report results using "% within 1%", the percentage of locations in a sample that are within 1% relative error of the true value, as this is more interpretable than a mean-square error.

**Results.** Figure 3 (left) shows results on the heat diffusion dataset for SMoEs and a number of baselines: CNNs, LCNs, fully-connected (FC) layers, CoordConv [65], CondConv [102], LRL-CNs [30], vision transformers (ViT) [26], and vision MoEs (V-MoE) [82]. For the LCN and FC layers, we use only a single layer as additional ones showed no benefit. Convolution, CoordConv, CondConv, and LRLCN are the best network found among a set with up to three layers, 3×3 kernels, 12 filters per layer, batchnorm, and ReLU activations. ViT and V-MoE use one transformer block with a patch size of 4×4, an embedding dimension of 128, and four heads. LRLCNs use three basis filters and an input-dependent combiner. We also tried unshared combining weights with no input dependence (LRLCN-ND). V-MoEs select one expert from a set of three. Our SMoEs use a single layer with three experts and select $E = 1$ expert per point. All models were trained with batch size 32, Adam [54] with a learning rate of 0.001 (decayed by 10× after no validation improvement for 15 epochs), and early stopping after no improvement for 30 epochs. Additional hyperparameter tuning did not significantly improve results. We report SMoE parameters as expert+gate parameters.

SMoEs achieve perfect performance on this dataset. Further, by examining the learned routing and experts (Fig. 3, right), we can see that it has indeed correctly learned the diffusion stencils and location-dependence. LCNs also achieve this, but require 3× more parameters, and require 110 epochs to converge (versus 8 for SMoEs). Fully-connected layers failed to learn the data well, likely due to the challenge of optimizing so many parameters. Other methods all converge to between 91 and 94% within 1%. Examining their predictions and weights, we observe that they do not appear to have learned the location-dependence of the diffusivities, and instead converged to predicting with an "average" diffusivity across the domain. We also tried larger (deeper and/or wider) convolutional networks, but performance did not improve. MoE methods (CondConv and V-MoE) also fail in this manner, as their coarse-grained experts are unable to specialize. Further, the LRLCN-ND fails in this manner, despite its architecture being similar to an SMoE when there is one output channel (a location-dependent, softmax-weighted combination of three basis kernels). We believe the LRLCN-ND exhibits a similar gradient "mismatch" as discussed earlier (§2.2).

We now discuss a number of different ablations of the SMoE architecture and design.

**What if the "right" expert configuration is not known?** While in the above experiments, we were able to select the SMoE expert configuration (number of experts, number of selected experts, expert filter size) so that it is both necessary and sufficient to learn the task, in many situations this information may not be available. We considered alternative SMoE configurations varying each of these parameters: ❶ using six experts; ❷ experts with 5×5 kernels; ❸ and selecting two experts per location from six total. For case ❸, we summed the two SMoE output channels together.

In all three cases, the SMoE achieved $100.0^{\pm 0}$% within 1% on the heat diffusion task. In ❶, we found that they learned duplicate diffusion stencils and still routed them appropriately. ❷ learned the five-point stencil plus a boundary of near-zero values, thus being nearly identical to the 3×3 kernel. Finally, ❸ learned diffusion stencils that summed together to produce the correct diffusivity. Thus, we can see that SMoEs are robust and adapt well to these sorts of architecture choices.

**RC loss and expert error damping.** Table 1 shows results for training SMoEs with and without our routing classification loss (§2.2) and expert error damping (§2.3). Without the RC loss, SMoE performance is at par with other baselines in Fig. 3, but once it is added, performance improves significantly as the gating function now learns the location-dependency in the data. Adding expert error damping further improves performance and convergence by limiting the impact of gate misroutings

Table 3: WeatherBench [78] results (latitude-weighted RMSE).

| Model | Z500 [m² s⁻²] | | T850 [K] | |
|---|---|---|---|---|
| | 3 days | 5 days | 3 days | 5 days |
| Rasp and Thuerey [80] | $316^{\pm 2.4}$ | $563^{\pm 3.1}$ | $1.80^{\pm 0.02}$ | $2.84^{\pm 0.03}$ |
| ➡ 2× wide | $310^{\pm 2.0}$ | $555^{\pm 2.8}$ | $1.76^{\pm 0.03}$ | $2.78^{\pm 0.01}$ |
| LRLCN [30] | $290^{\pm 1.4}$ | $549^{\pm 1.9}$ | $1.73^{\pm 0.03}$ | $2.79^{\pm 0.01}$ |
| ViT (2×2) [26] | $438^{\pm 2.8}$ | $638^{\pm 3.1}$ | $2.24^{\pm 0.04}$ | $2.88^{\pm 0.03}$ |
| SMoE after first layer | $305^{\pm 1.9}$ | $556^{\pm 2.2}$ | $1.77^{\pm 0.01}$ | $2.80^{\pm 0.03}$ |
| Last layer SMoE | $298^{\pm 2.6}$ | $553^{\pm 3.2}$ | $1.73^{\pm 0.02}$ | $2.78^{\pm 0.04}$ |
| 3×3 convs→SMoE | $278^{\pm 2.0}$ | $530^{\pm 1.8}$ | $1.69^{\pm 0.01}$ | $2.65^{\pm 0.01}$ |
| ➡ + gate prior | $\mathbf{270}^{\pm 1.9}$ | $\mathbf{525}^{\pm 2.0}$ | $\mathbf{1.66}^{\pm 0.02}$ | $\mathbf{2.60}^{\pm 0.01}$ |
| ➡ rand fixed gate init | $328^{\pm 3.7}$ | $572^{\pm 4.1}$ | $1.89^{\pm 0.08}$ | $2.96^{\pm 0.05}$ |
| R&T [80] (pretrained) | $267^{\pm 1.8}$ | $500^{\pm 2.4}$ | $1.66^{\pm 0.03}$ | $2.43^{\pm 0.02}$ |
| SMoE (pretrained) | $253^{\pm 2.1}$ | $488^{\pm 1.7}$ | $1.57^{\pm 0.02}$ | $2.34^{\pm 0.02}$ |
| ➡ + extra ERA5 | $232^{\pm 1.5}$ | $440^{\pm 1.2}$ | $1.46^{\pm 0.02}$ | $2.19^{\pm 0.01}$ |
| ➡ + 1.4° | $\mathbf{198}^{\pm 1.8}$ | $\mathbf{382}^{\pm 2.0}$ | $\mathbf{1.42}^{\pm 0.00}$ | $\mathbf{2.06}^{\pm 0.02}$ |

Table 4: ImageNet [24] validation accuracy.

| Model | Top–1 % | Top–5 % |
|---|---|---|
| ResNet-50 [42, 94] | $80.83^{\pm 0.04}$ | $95.39^{\pm 0.03}$ |
| LRLCN [30] | $80.90^{\pm 0.02}$ | $95.41^{\pm 0.05}$ |
| SMoE after first layer | $80.85^{\pm 0.05}$ | $95.40^{\pm 0.01}$ |
| Last layer SMoE | $80.91^{\pm 0.04}$ | $95.42^{\pm 0.03}$ |
| 3×3 convs→SMoE | $81.33^{\pm 0.03}$ | $95.52^{\pm 0.01}$ |
| Wide ResNet-50-2 [103] | $\mathbf{81.76}^{\pm 0.03}$ | $\mathbf{95.74}^{\pm 0.02}$ |

on expert learning. However, damping on its own offers little benefit, as it does not improve gate learning. These results show that these refinements are critical for good performance.

**Gating function.** Table 2 shows the performance of different gating functions (§2.1) on the SMoE. We consider six options: A single fully-connected layer (as is commonly used in MoEs [82, 86]); a single 3×3 convolution, CoordConv [65], or LCN layer; a gate with three CoordConv layers with batchnorm and ReLU; and our tensor routing gate. When training, we also considered auxiliary losses and other methods for improving performance (see §C) and report the best result. Our tensor routing offers the best performance. An LCN performs second-best, likely because it also uses separate parameters per location, but uses 9× as many parameters and requires significant computation. Other methods do not appear able to effectively capture location-dependence.

**Other ablations.** We conduct a number of additional ablation studies in §C, including using auxiliary losses and routing noise during training, routing normalizations, and expert functions.

## 3.2 Medium-Range Weather Prediction

We now discuss results on the WeatherBench [78] medium-range weather forecasting benchmark. This benchmark uses the ERA5 reanalysis dataset [44], with hourly global weather data for 1979–2018. We use the data subset suggested by Rasp et al. [78] at 5.625° resolution (32×64 grid points) and train on data from 1979–2015, validate on 2016, and report test results for 2017–2018. We otherwise follow the training methodology of Rasp and Thuerey [80]. The target quantities to predict are geopotential at 500 hPa (Z500) and temperature at 850 hPa (T850) with a three- and five-day lead time. We report results using latitude-weighted root-mean-square error (RMSE).

As a baseline, we use the ResNet architecture [42] introduced by Rasp and Thuerey [80], which currently reports the best results on WeatherBench. This architecture consists of 19 residual blocks each with two [3×3 convolution → LeakyReLU → batchnorm → dropout] layers, plus an initial 7×7 convolution layer. All convolutions but the last have 128 filters. We consider three additional baselines. The first is identical to the above, but with twice as many filters (256) in each convolution. Second, we replace 3×3 convolutions with LRLCN [30] layers. Finally, we use a four-layer ViT [26] with patch size 2×2, hidden dimension 1024, and eight heads (the best performing configuration).

We adapt the Rasp and Thuerey ResNet to use SMoEs with three configurations: adding an SMoE layer after the first convolution; adding an SMoE layer after the final convolution; and replacing all 3×3 convolutions with SMoE layers. Each SMoE selects the same number of experts as the original layer had filters, and has twice as many experts (i.e., $|\mathcal{E}| = 256$, $E = 128$). We also *share* the tensor routing gate across all SMoE layers with the same number of experts, so its overhead is minimal.

Because the weather data is on a fixed grid with underlying location-dependence (the Earth), we expect SMoEs to convey some benefit by specializing to the characteristics of different regions. In Table 3, we observe that this is indeed the case. Adding SMoEs improves results in all situations, with the most significant improvement coming through replacing all 3×3 convolutions with SMoEs. This showcases the advantage of incorporating appropriate location-dependent biases. Wider ResNets

Table 5: Results for prediction correction on the ENS-10 [8] dataset for ensemble post-processing.

| Metric | Model | Z500 [m$^2$ s$^{-2}$] | | T850 [K] | | T2m [K] | |
|---|---|---|---|---|---|---|---|
| | | 5-ENS | 10-ENS | 5-ENS | 10-ENS | 5-ENS | 10-ENS |
| CRPS | EMOS | 79.12$^{\pm 0.12}$ | 78.80$^{\pm 0.21}$ | 0.721$^{\pm 0.01}$ | 0.706$^{\pm 0.04}$ | 0.720$^{\pm 0.00}$ | 0.711$^{\pm 0.03}$ |
| | U-Net | 76.54$^{\pm 0.20}$ | 76.18$^{\pm 0.12}$ | 0.685$^{\pm 0.00}$ | 0.670$^{\pm 0.01}$ | 0.657$^{\pm 0.01}$ | 0.644$^{\pm 0.01}$ |
| | SMoE | **68.94**$^{\pm 0.14}$ | **67.43**$^{\pm 0.12}$ | **0.612**$^{\pm 0.01}$ | **0.590**$^{\pm 0.02}$ | **0.601**$^{\pm 0.02}$ | **0.594**$^{\pm 0.02}$ |
| EECRPS | EMOS | 29.21$^{\pm 0.18}$ | 29.02$^{\pm 0.13}$ | 0.247$^{\pm 0.00}$ | 0.245$^{\pm 0.02}$ | 0.244$^{\pm 0.00}$ | 0.241$^{\pm 0.02}$ |
| | U-Net | 27.78$^{\pm 0.11}$ | 27.55$^{\pm 0.19}$ | 0.230$^{\pm 0.01}$ | 0.229$^{\pm 0.01}$ | 0.225$^{\pm 0.00}$ | 0.220$^{\pm 0.01}$ |
| | SMoE | **23.79**$^{\pm 0.20}$ | **23.10**$^{\pm 0.16}$ | **0.207**$^{\pm 0.03}$ | **0.197**$^{\pm 0.03}$ | **0.199**$^{\pm 0.01}$ | **0.190**$^{\pm 0.02}$ |

offer limited improvement (in line with results reported by Rasp and Thuerey [80]). The location-dependent filters of LRLCNs improve over ResNets, but fail to match SMoEs. We were unable to achieve good performance with ViTs, but did observe that they are highly sensitive to patch size.

**Incorporating prior knowledge into gates.** While the exact nature of the location-dependence of this data is unknown, we do have a broad prior on some aspects of it, such as whether a point is land or sea. This information can be incorporated into an SMoE by initializing the tensor routing gate to bias routing to different experts. To this end, we use the land-sea mask from ERA5 to initialize the gate to route land locations to half the experts and sea locations to the other half. Note this does not fix the routing, as the gate is able to adjust as it learns. Further, the land-sea mask is already included in the input data, so all models already had access to this information.

Results with this are in the "+ gate prior" line of Table 3, and perform best. This configuration sets a new state-of-the-art for WeatherBench when not using additional data. Indeed, it nearly matches the performance of a ResNet with 150 years of additional pretraining data from climate simulations [80].

We also tried a configuration where the gate was initialized randomly and fixed rather than learned ("rand fixed gate init"). This performs worse than our baseline, as the network cannot adapt its routing choices, and each expert sees even fewer points in each sample than a standard network, resulting in less learning. Thus, learning the routing function is critically important to good performance.

**Additional data.** Following Rasp and Thuerey [80], we use 150 years of data from the MPI-ESSM-HR climate model from the CMIP6 archive [32] to pretrain our best SMoE configuration, which was then fine-tuned on ERA5 data as above. This significantly outperforms both our SMoEs without pretraining and Rasp and Thuerey's pretrained ResNet. We incorporated more data to further push the performance by adding ERA5 data from the most recent back extension (1959–1979), increasing the dataset size by about 50%. This shows improved results; however, we suspect performance is saturating due to the coarse spatial resolution of the data. We therefore trained a final configuration with higher resolution (1.4°) data. Using this, our SMoEs *significantly outperform the state-of-the-art on WeatherBench*; indeed, its performance on T850 is very close to that of the operational Integrated Forecast System [78]. Our results are also competitive with those of Keisler [53], although they are not directly comparable (due to, e.g., different data resolutions).

**Mode collapse.** Many MoEs suffer from expert or mode collapse (e.g., [10, 82, 86]), where only a small number of experts are selected. This is typically avoided with routing noise and/or auxiliary "load balance" losses. On the heat diffusion dataset, we found these losses to offer no benefit (§C). We also did not observe mode collapse in SMoEs on WeatherBench. With the RC loss, we directly train the gate, updating routing weights toward other experts after mistakes, and so avoid such issues.

**Expert selection order.** During training, the order experts are concatenated may change (due to changes in relative routing scores, or selecting different experts), which will impact the order of channels seen by subsequent layers. When training on WeatherBench, we found this not to have a significant impact: expert order stabilizes early, allowing layers to operate on stable representations. Further, most "swapping" occurs among low-confidence experts, so is limited to a subset of channels.

## 3.3 Post-Processing Ensemble Weather Forecasts

Numerical weather prediction systems typically utilize ensembles of simulations in order to quantify uncertainty and improve forecast quality [14]. However, such ensembles typically exhibit systematic biases [92], and correcting them improves forecast skill [14, 85, 101], a task for which deep learning has shown promise [38, 79]. We use the ENS-10 dataset [8], which consists of twenty years (1998–

2017) of global reforecast [40] data at $0.5°$ spatial resolution. We follow the benchmarking setup of Ashkboos et al. [8], and correct predictions for Z500, T850, and 2 meter temperature (T2m) at a 48 hour lead-time using both five and ten ensemble members. We report results using continuous ranked probability score (CRPS) and extreme event weighted CRPS (EECRPS).

We adapt the U-Net model from the ENS-10 baselines, as it delivers good performance and operates on global data (other methods use patches). Similar to our approach for WeatherBench, we replace each 3×3 convolution with an SMoE with four times as many experts as the original layer, and select the same number of experts as the original layer had filters. We share tensor routing gates between all layers with the same spatial dimensions and number of experts, with the exception that the encoder and decoder trunks also use separate gates. As baselines, we use the original U-Net architecture and Ensemble Model Output Statistics (EMOS) [36], a standard post-processing method.

We observe in Table 5 that, similar to WeatherBench, SMoEs offer significant improvements in forecast skill across all situations, and set a new state-of-the-art for prediction correction on the ENS-10 dataset. This also demonstrates that SMoEs are able to scale to the very large spatial domain used by the ENS-10 data and still learn the appropriate location dependence.

### 3.4 Image Classification

Lastly, we present results on several image classification tasks; we focus here on ImageNet-1k [24] and discuss results on additional datasets in §D. While these datasets do not have a strict location-dependent structure, relaxing the strict translation equivariance of convolutions can bring benefits, and enables a direct comparison with Elsayed et al. [30]. We follow their experimental methodology and train using the recipe of Vryniotis [94]. We either insert an SMoE layer after the first or last convolutional layer of ResNet-50 [42] or replace all 3×3 convolutions with SMoE layers. Our SMoEs have twice as many experts as the original convolution layer and select half of them, to keep output dimensions constant. Gating layers are shared among all equally-sized blocks. For comparison, we also train ResNet-50 with all 3×3 convolutions replaced by LRLCN [30] layers; and a Wide ResNet-50-2 [103], which has comparable parameters to SMoEs.

Table 4 shows that SMoEs outperform LRLCNs when we replace all 3×3 convolutions, while using 56% of the parameters. However, a wide ResNet performs best overall. Nevertheless, this shows that ImageNet classification does indeed benefit from relaxing translation equivariance.

## 4 Discussion

We presented the Spatial Mixture-of-Experts layer, a novel layer that learns underlying location dependencies in data and then uses fine-grained routing to specialize experts to different areas. We also introduce a routing classification loss and expert error damping, which enable SMoEs to perform well on dense regression tasks. Prior work shows limited effectiveness on these tasks: Either it does not capture location-dependence (e.g., convolutions) or it operates at a coarse-grained level (e.g., standard MoEs). By overcoming these challenges, we show a new capability for neural networks, and set new state-of-the-arts for medium-range weather prediction and ensemble post-processing.

Many other problems of broad societal import have a similar spatial structure, particularly in scientific domains [9], and we expect SMoEs to be applicable to them. However, tasks such as facial recognition and surveillance have also historically shown benefit from such improvements [91] and SMoEs should therefore be used with care.

SMoEs show that learning location-dependence is a powerful inductive bias for certain types of data, and there are many avenues for further study. Two key areas of particular interest are to develop improved implementations for fine-grained, sparse routing; and to generalize SMoEs from operating on grids to general graphs, which would enable them to be applied to many additional tasks.

**Acknowledgements and Disclosure of Funding**

We thank the members of SPCL at ETH Zürich 🏔️, and Peter Dueben and Mat Chantry of ECMWF, for helpful discussions; and the anonymous reviewers for their suggestions and feedback. This work has received funding from the European High-Performance Computing Joint Undertaking (JU) under grant agreement No. 955513 (MAELSTROM), and from Huawei. N.D. received support from the ETH Postdoctoral Fellowship. We thank the Swiss National Supercomputing Center (CSCS) and Livermore Computing for computing infrastructure.

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
