# OpenReview forum: "Spatial Mixture-of-Experts"
_NeurIPS.cc/2022/Conference — NeurIPS 2022 Accept_

### Official Review · Reviewer_Y3eL · 2022-07-08

**Rating:** 5
**Confidence:** 3
**Soundness:** 2 fair
**Presentation:** 3 good
**Contribution:** 2 fair

**Summary:**

The authors propose a convolutional neural architecture that learns parameters specialized towards different locations in the input image. This can be seen as a middle way between a ConvNet, where all parameters are shared across spatial locations and fully-connected networks, where all parameters are specific for their location. Specifically, the authors propose a learnable matrix that assigns each spatial location to a subset of convolutional filters. To learn this matrix, the authors propose a routing classification loss that aims to change assignments for locations with large gradients. The method is then evaluated on a small set of specialized datasets, where it performs well.

**Questions:**

Would the authors be able to highlight a series of key tasks where such a model could provide impact?

**Limitations:**

As far as I can see, the authors have not addressed potential limitations. However, I do not see any clear issues.

**Strengths And Weaknesses:**

Making neural models larger has been an effective way to improve model performance. Making those large models sparse has been shown to be a successful way to combine large capacity models with fast (theoretical) inference and training speeds. As such the paper at hand is of interest to the research community.

Overall the paper is well presented and the method is clearly explained.

One concern is that the authors make a very strong assumption in that SMOE relies on the spatial structure to be similar across all samples in a dataset. This leads to very limited applications and no generalization across datasets. Given this very large limitation, it would have been great if the authors could extend their motivation and highlight a series of key tasks where such a model could provide impact.

---

> ### Author Response · Authors · 2022-08-02
> **Initial Response**
>
> Thank you for the suggestions. We have adjusted the paper accordingly.
>
> > One concern is that the authors make a very strong assumption in that SMOE relies on the spatial structure to be similar across all samples in a dataset … Would the authors be able to highlight a series of key tasks where such a model could provide impact?
>
> This is indeed a strong assumption, but also one that enables SMoEs to provide their inductive bias and consequent performance improvement when it does apply.
>
> We see several sets of tasks where SMoEs could provide impact:
> - Tasks involving faces (e.g., facial recognition or face landmarks) commonly have such spatial structure.
> - There are a number of tasks within weather and climate where incorporating spatial structure may have benefit, beyond the weather prediction results on WeatherBench we show. For example, post-processing numerical weather prediction ensembles can benefit from locality (e.g., Grönquist et al. incorporates a LCN for bias correction for this reason), as could data assimilation pipelines. Indeed, many country- or locality-level weather services refine global weather model outputs with fine-grained local information.
> - Many scientific numerical simulations are run on a fixed underlying geometry where spatial information could be key.
> - More generally, for many tasks, this spatial structure may hold approximately, to the point that relaxing strict translation equivariance offers performance improvements. Alternatively, data can often be registered or aligned in advance as part of a preprocessing pipeline.
>
> > As far as I can see, the authors have not addressed potential limitations. However, I do not see any clear issues.
>
> We appreciate that you found no clear issues. We see two key limitations of SMoEs (discussed in Sec. 1 and 2): For SMoEs to be applicable, data need to share the same (or similar) underlying spatial structure; and, the current implementation of SMoEs is limited to data on Cartesian grids.

---

### Official Review · Reviewer_vuZ5 · 2022-07-09

**Rating:** 6
**Confidence:** 4
**Soundness:** 2 fair
**Presentation:** 3 good
**Contribution:** 3 good

**Summary:**

This paper presents a location-based mixture of experts for use in applications where modeling depends on the image grid location, such as weather forecasting.  Experts are selected using a map of size (E,H,W), where E is the size of experts pool, so that one set of top-k experts are learned for each grid location.  In order to encourage the system to select appropriate experts, an auxiliary loss is added that classifies whether the selected experts are "correct" or "incorrect" based on quantile of the gradient magnitude; experts with large error gradient are assigned a uniform classification target among the other experts, encouraging the system to find a new one.  The largest gradients are also scaled down in their bprop application, with the intuition that they will be applied to experts that may no longer be selected.  The system is verified on a toy dataset constructed from synthesized heat diffusion, and evaluated on WeatherBench and Imagenet, obtaining significant improvements in weather forecasting.


**Questions:**

Additional questions:

l.194:  The claim that sparse selection results in the same number of FLOPS as a standard conv layer with same number of filters, is not quite right.  For each location, the additional patch context must also be included.  In the worst case, each neighboring location selects different experts in a 3x3 checkerboard pattern, in which case FLOPS required would be that of a standard convnet with the same number of filters times the spatial kernel size (e.g. 9).  And even this depends on having a sparse implementation when there are more than 9 experts to select from.

Fig 3:  do you have any ideas on why LRLCN-ND performs worse than SMoE on the toy example? it's a very similar architecture for 1 output channel, the only difference is that it uses per-location softmax mixture of the 3 kernels, rather than top1 max.  will adding RC loss fix this?

If land and water are in different locations between images, is it possible to feed the land/water map in as a binary map and gate based on this, rather than location?


**Limitations:**

-

**Strengths And Weaknesses:**

The idea of incorporating location dependence for some applications, like weather forecasting, is sound, and well-suggested by the results in this paper.  Recovery of near-exact diffusion kernels in the toy data is good to see.  However, there are some additional baseline comparisons I think are missing, in particular, LCN, LRLCN and resnet with wider layers to match the number of parameters of SMoE on WeatherBench.  SMoE with a fixed set of random 128/256 experts at each location selected, as in the init, without learning the gating map, would also be interesting, as improvement over this would show the effect of learning the map in WeatherBench.  Measuring on an additional relevant application, such as face landmarks or another of the ones mentioned in the intro, would also strengthen the demonstrated reach of the method.

There are also some details around the order of the channels (experts) selected, and if that matters.  Are the top ones per location selected in index order, or sorted gate score order?  When a filter changes, and an expert i is replaced with expert j, are experts with indices between i and j will shifted in the output representation happens?  This would look like a sudden shift permutation to the next layer.  And if experts are selected in score order, then as scores change, input channels to the next layer will swap when scores change during training.

As an aside, I think it's debatable whether the system described is a mixture of experts.  It seems more to be a channel subselection, since the outputs are concatenated instead of mixed, and subselection is not dependent on the same input as the layers.  This is a very minor point, though, it seems close enough to me to be an accurate description.

Overall, this is an interesting idea, and one that clearly seems effective at incorporating effects of spatial differences, such as land vs water in weather prediction.  A few comparisons appear to be missing in the experiments (described above), and while the toy heat diffusion experiment and existing ablations serve to profile the behavior of the system, I still would have liked to see more discussion on the effects of discontinuous subselection to downstream layers.

---

> ### Author Response · Authors · 2022-08-02
> **Initial Response**
>
> Thank you for the detailed feedback. We have made updates to the paper accordingly.
>
> > in particular, LCN, LRLCN and resnet with wider layers to match the number of parameters of SMoE on WeatherBench
>
> This is a good suggestion, and we have added additional experiments (please see our general comment). We did not run an LCN baseline due to its large memory and training time requirements.
>
> > MoE with a fixed set of random 128/256 experts at each location selected, as in the init, without learning the gating map, would also be interesting
>
> Thank you for the suggestion — we have added an experiment in this setting, showing the advantage of learning the gating map. (See our general comment.)
>
> > Measuring on an additional relevant application, such as face landmarks or another of the ones mentioned in the intro, would also strengthen the demonstrated reach of the method.
>
> We agree with this. While we unfortunately don’t believe that we can get results for an additional application within the rebuttal period, we plan to add them to the final version of the paper.
>
> > There are also some details around the order of the channels (experts) selected, and if that matters.
>
> Experts are concatenated in sorted score order (based on the indices produced by `torch.topk`). This does indeed mean that the order of channels can change during training, but empirically we observe that this stabilizes quite early during the training process. Most "swapping" seems to occur with selected experts that a gate has low confidence in, so even when this does occur it is limited to a subset of the channels. We have added a short discussion of this to the paper in Secs. 2.4 and 3.2.
>
> > The claim that sparse selection results in the same number of FLOPS as a standard conv layer with same number of filters, is not quite right.
>
> We believe this reasoning is not quite correct, and the claim in our paper stands. To clarify: Suppose we have a SMoE layer that selects E experts at each point, and a convolutional layer with E filters, and the layers are otherwise identical (same kernel size, etc.). The convolution will apply the E filters at each point, for a total compute cost of O(ECHWK2) (C input channels channels, height H, width W, and kernel size K, neglecting stride and assuming "same" padding). An SMoE will select E filters to apply at each point, which are then applied as if they are normal convolution filters (and use the same context surrounding context, etc.). Hence, the flop cost is the same, it's just that an SMoE applies a different set of filters at each point.
>
> > do you have any ideas on why LRLCN-ND performs worse than SMoE on the toy example? … will adding RC loss fix this?
>
> We believe this is an issue with using an LRLCN for such a regression task, where there is a gradient "mismatch" similar to what we observed with SMoEs (Sec. 2.2). Indeed, in our experiments, using softmax for SMoE routing normalization, without the RC loss and expert error damping, offered no performance improvement (with performance similar to that case in Table 1). Reformulating the LRLCN to be trained with the RC loss may indeed improve its performance.
>
> > If land and water are in different locations between images, is it possible to feed the land/water map in as a binary map and gate based on this, rather than location?
>
> This is possible. We considered including a small network as part of the gate that learned to incorporate additional information, but found it did not offer additional improvements on the heat diffusion task, although it may be more useful for more complicated tasks like WeatherBench. It would have the downside of adding compute to the gating function.

---

> > ### Comment · Reviewer_vuZ5 · 2022-08-06
> > **response**
> >
> > Thanks for your responses.  In particular, I think the new experiments with wider resnet and LRLCN on WeatherBench improve the comparisons.  I've updated my score to 6.
> >
> > I also now agree with the claim that FLOPs are the same as a conv layer with the same kernel size, assuming sparse implementation --- I think I had included the kernel size differently before, somehow thinking of channel selection instead (thanks for the clarification on this).
> >
> > Reading the other reviews, I see one concern is the method's reliance on constant spatial alignment between samples.  I noticed this as well, but do not view it as a major concern for this work.  While of course allowing more variability would be an improvement, as the authors point out, there are applications where near-exact alignment or a registration preprocessing step are reasonable.
> >
> > If there are measurements with improvement on an additional relevant application, even if not SotA but compared to strong enough appropriate baseline, IMO that would empirically demonstrate a greater reach and bring this from borderline/weak accept to clear accept.

---

> > > ### Author Response · Authors · 2022-08-09
> > > **Thanks**
> > >
> > > Thank you!
> > >
> > > > If there are measurements with improvement on an additional relevant application, even if not SotA but compared to strong enough appropriate baseline, IMO that would empirically demonstrate a greater reach and bring this from borderline/weak accept to clear accept.
> > >
> > > We are working on additional experiments, which would be ready for the final version of the paper. We plan to show results on an additional weather task (ensemble post-processing) and on a separate task.

---

### Official Review · Reviewer_cLKX · 2022-07-11

**Rating:** 6
**Confidence:** 3
**Soundness:** 4 excellent
**Presentation:** 3 good
**Contribution:** 3 good

**Summary:**

This paper proposes a new way of tailoring the processing of subparts of the input data based on their localisation, in the case where all samples from a given dataset share the same spatial structure. This can be of use in cases where treating the localisation as a mere input feature is not enough, but appeals for more drastic measures breaking the spatial invariance properties of some classic networks such as CNN.

The method presented relies on learning different experts, each being specialized in processing certain parts of the input data. At each localisation, the choice of experts to use is done by looking at a "routing table", scoring each expert for each localisation, regardless of the input sample. There is only one routing table for the whole dataset: the routing is independent of the input sample. It is learnt via a newly introduced method treating the routing as a classification task, with labels derived from the loss of the task at hand.

The method is shown to compare favorably to previous baselines on a task created for this paper where the "true parameters" of the model generating the sythetic data are known, so as to perform an in-depth ablation study highlighting the impact of each design decision. On a weather-prediction benchmark using real-world data, the method is shown to be state-of the art, and can also bring a slight improvement to a ResNet 50 on Imagenet.

**Questions:**

* On the heat diffusion dataset, did you try to compare yourselves to some baseline CNN architectures having the same number of parameters  as your SMoE model ? (In Figure 3, I only see a line writing 146 for the CNN vs 27+12288 for SMoE)
* Is it possible to compare SMoE to the other methods on WeatherBench ?
* The quantile of exactly which distribution is used to create the labels in the training of your routing tensor ?
* In the practical implementation part, you said that all experts at all points are evaluated. If I am understanding right, this means that in the last layer of your model, instead of using the $\texttt{scatter}$ and $\texttt{gather}$ functions of Pytorch, you could directly evaluate the loss for all experts. Would it be useful to make use of this signal to train your routing system ?

* (there is a small typo line 311: "Each SMOE ~~has~~ selects")

**Limitations:**

Yes, the authors adressed the limitations and potential negative societal impact in their discussion paragraph.

**Strengths And Weaknesses:**

$\textbf{Strengths}$:

The paper is clearly motivated and well written overall. The method introduced to train both the routing system and the experts at the same time is original and kept simple, the relevance of each component is thoroughly tested, and the robustness of the model to changes in variables unknown by the user is challenged in a controlled setting on synthetic data. The method is shown to work in practice on real-world data.

$\textbf{Weaknesses}$:

* The explanation of the procedure used to create the labels for the classification task to train the routing tensor is a bit unclear to me: the quantile of exactly which distribution is used to create the labels ?
* In WeatherBench (Table 3), SMoE is only compared to one method whereas other models leveraging the position where tested and shown to be on par with SMoE performances on the heat diffusion data (Figure 3.a) and simple Image datasets (Table D.1 and D.2). It would be interesting to know how those previously introduced methods fare on a case where it seems breaking the spatial invariance makes sense.

---

> ### Author Response · Authors · 2022-08-02
> **Initial Response**
>
> Thank you for the detailed comments and suggestions. We have updated the paper accordingly.
>
> > The explanation of the procedure used to create the labels for the classification task to train the routing tensor is a bit unclear to me: the quantile of exactly which distribution is used to create the labels ?
>
> We use the distribution of the error signal to the layer (i.e., the gradient of the layer's output).
>
> > In WeatherBench (Table 3), SMoE is only compared to one method … It would be interesting to know how those previously introduced methods fare on a case where it seems breaking the spatial invariance makes sense.
>
> This is a good suggestion, and we have added additional experiments (please see our general comment).
>
> > On the heat diffusion dataset, did you try to compare yourselves to some baseline CNN architectures having the same number of parameters as your SMoE model?
>
> Our CNN architectures report the best result from a set of networks with up to three layers and twelve filters per layer. We manually tried even larger (wider and/or deeper) networks, but they did not offer improvements.
>
> > In the practical implementation part, you said that all experts at all points are evaluated. If I am understanding right, this means that in the last layer of your model, instead of using the scatter and gather functions of Pytorch, you could directly evaluate the loss for all experts. Would it be useful to make use of this signal to train your routing system ?
>
> This would indeed be possible with our current implementation; however, it would not be possible with a more optimized implementation that only evaluates the selected experts at the points they are applied at. For that reason, we don’t want to incorporate information that would require a different layer implementation. Additionally, while this could be useful for the final layer, it may be less useful for intermediate errors.
>
> > (there is a small typo line 311: "Each SMOE ~has~ selects")
>
> Thanks for pointing this out!

---

### Official Review · Reviewer_811C · 2022-07-12

**Rating:** 7
**Confidence:** 4
**Soundness:** 3 good
**Presentation:** 3 good
**Contribution:** 3 good

**Summary:**

The paper presents a method based on spatial MoE for location-dependent learning. Unlike the translation-equivariant network (i.e., CNN) that shares the expert (convolution kernel) locally, the proposed method assigns the convolution kernel conditional on the location. Specifically, the work proposes a novel gating function to select an expert for each location and use the expert to extract information for the location. The paper validates the effectiveness of their method in multiple tasks and shows the SOTA results on medium-range weather prediction.

**Questions:**

Does the gating learn the translation-equivariance? It might be interesting to look at. The reason I ask is that, for location-dependent earth data, I still believe that the translation-equivariance could be helpful since patterns are repetitive. And as shown in Fig.3, I saw that there is a local pattern (e.g., repetitive local patches) in the expert selection.

**Ethics Review Area:**

["I don’t know"]

**Limitations:**

The author addressed the limitations.

**Strengths And Weaknesses:**

Originality:
The paper presents a novel viewpoint on utilizing deep networks. For some tasks, it's beneficial to violate the translation-equivariance. This is very interesting to the research community. Furthermore, the proposed method is also novel. The proposed gating function and classification loss obviate the need for a variety of regularizers.

Quality:
The method supports the claims well with multiple experiments. The only concern I have is the missing analysis of mode collapse -- a typical pitfall of MoE. For completeness, it would be better to give more analysis -- for example, is there any mode collapse and any insight/discussion on why the proposed method can avoid it?

Clarity:
The paper is well-written in general. I only have minor comments.
- The motivation behind the proposed tricks is missing. The paper didn't show the logic behind the proposed tricks. For example, why is the expert error damping essential, and the logic behind its design?

Significance:
The method is novel. The results are also strong and very interesting -- by violating translation-equivariance, the method shows improved performance.

---

> ### Author Response · Authors · 2022-08-02
> **Initial Response**
>
> Thank you for the comments and suggestions! We have updated the paper with additional discussion and respond below.
>
> > The only concern I have is the missing analysis of mode collapse -- a typical pitfall of MoE. For completeness, it would be better to give more analysis -- for example, is there any mode collapse and any insight/discussion on why the proposed method can avoid it?
>
> This is a great question. In our initial explorations with SMoEs on the heat diffusion dataset, we did indeed observe degenerate cases where either the gate routed to a single expert everywhere or simply never changed its routing decisions from those made at initialization. The standard trick for avoiding this is a combination of routing noise and auxiliary losses (e.g., as in Shazeer et al.). We also tried these, and found they did not offer any significant performance improvement. (Tab. C.2 in the supplementary reports studies using auxiliary losses _with_ the routing classification loss and expert error damping; we have added an additional table showing performance when not using them.)
>
> We found that by using the routing classification loss, issues with mode collapse were entirely avoided. We believe this is because the separate loss function provides a better signal to the gate, allowing it to better penalize bad routing decisions at a location while encouraging the gate to route to other experts, and so helps avoid getting stuck in bad local minima. We added a discussion of this to Sec. 3.2.
>
> > The motivation behind the proposed tricks is missing. The paper didn't show the logic behind the proposed tricks. For example, why is the expert error damping essential, and the logic behind its design?
>
> We discuss some motivation behind the routing classification loss and expert error damping in Sec. 2.2 and 2.3, and provide further motivation in Appendix B. We have expanded our discussion in Sec. 2.2 and 2.3.
> - In short, we developed the RC loss because of an observed "gradient mismatch". When training end-to-end with an MSE loss for regression, gradients may be positive or negative simply because of what the true value should be at a point (e.g., a prediction was positive, but should be a smaller value). This same gradient would be used to update the gate, but a negative gradient there may incorrectly inhibit routing to an expert.
> - Expert error damping has a similar motivation. If the gate routes a location to an expert incorrectly in an iteration, that expert will be updated to push its output toward a better prediction at that location, despite the fact that the update may make the expert worse at locations that are correctly routed to it. Indeed, it may take multiple iterations for the gate to make enough updates to change its routing decision.
>
> > Does the gating learn the translation-equivariance? It might be interesting to look at.
>
> This is a quite interesting question. If the direct tensor routing learns translation-equivariance, it should route the same experts to every location. We performed a quick examination of a learned gate for CIFAR-10, and found that this is generally the case: most locations route to the same set of experts, and some experts are applied at (nearly) every location. However, there is still some variation here, so the gating is not exactly translation equivariant.

---

### Official Review · Reviewer_wHKK · 2022-07-14

**Rating:** 3
**Confidence:** 4
**Soundness:** 2 fair
**Presentation:** 2 fair
**Contribution:** 2 fair

**Summary:**

Authors proposed a spatial mixture-of-expert layer for CNNs to take into account of spatial dependence in data or features.

**Questions:**

Computational costs should be analyzed and compared.

**Limitations:**

Okay

**Strengths And Weaknesses:**

Strengths: to incorporate local dependence or locality in data and features is always beneficial. Yes, most CNNs do not explicitly consider this or have mechanisms to embed this feature into the learning process.

Weaknesses: authors create an impression that the proposed scheme is the only one that consider locality in data. There are always many mechanisms to incorporate local dependences in CNNs such as graphs, CRF or MRF, in segmentation and recognition. The proposed scheme seems only marginally improve performances if any, while added computational costs may outweigh the benefit.

Title is misleading - word "Layer" should be added.

---

> ### Author Response · Authors · 2022-08-02
> **Initial Response**
>
> Thank you for the comments.
>
> > authors create an impression that the proposed scheme is the only one that consider locality in data. There are always many mechanisms to incorporate local dependences in CNNs such as graphs, CRF or MRF, in segmentation and recognition
>
> We do not mean to give the impression that SMoEs are the only scheme to incorporate locality in data! Indeed, we discuss related work on incorporating local dependencies in the first paragraph of Sec. 1, Fig. 1, and the related work (Sec. 1.1, "Local spatial structure"). We have expanded our related work to incorporate discussion on additional mechanisms. Thank you for pointing these out. If there are additional key references you would like to see discussed, please let us know.
>
> > The proposed scheme seems only marginally improve performances if any, while added computational costs may outweigh the benefit.
>
> We respectfully disagree with this characterization. In particular, we achieve significant performance improvements on WeatherBench (Tab. 3), where, for example, we reduce latitude-weighted RMSE for Z500 at a 3-day lead time by 14.5%. Our SMoEs outperform the state-of-the-art on WeatherBench without using additional data. We also show and analyze significant improvements over a variety of baselines on our heat diffusion benchmark (Fig. 3, left), where SMoEs achieve perfect performance (100 % within 1%) and the next-best model achieves 93.8 % within 1% (excluding the LCN model, which uses many more parameters).
>
> > Computational costs should be analyzed and compared.
>
> We discuss computational costs in Sec. 2.4. In brief, the flop cost of an SMoE is the cost of the gate plus the cost of applying the experts. Using our tensor routing, the gate requires no flops (but does require some amount of memory to store). Applying convolutional experts (as in our formulation in Sec. 2) requires the same flops as a standard convolutional layer with a number of filters equal to the number of selected experts in the SMoE. We are happy to provide additional discussion if there are particular points you feel should be expanded.

---

### Author Response · Authors · 2022-08-02
**General Response**

We thank the reviewers for the detailed comments and suggestions on how to improve the paper. We appreciate that reviewers found the work novel and interesting, well-presented, and that it had strong results.

We have updated the manuscript based on the feedback here. We respond briefly to a common point below, and to specific reviewer questions in individual replies.

A common request was for results on WeatherBench with additional models (cLKX, vuZ5).

We have expanded our experiments on WeatherBench with a wider ResNet, a LRLCN, and a vision transformer. Results are below:

| Model                    | Z500 (3 days) | Z500 (5 days) | T850 (3 days) | T850 (5 days) |
| ------------------------ | ------------- | ------------- | ------------- | ------------- |
| Rasp & Thuerey           | 316±2.4       | 563±3.1       | 1.80±0.02     | 2.84±0.03     |
| Rasp & Thuerey 2x wide   | 310±2.0       | 555±2.8       | 1.76±0.03     | 2.78±0.01     |
| LRLCN                    | 290±1.4       | 549±1.9       | 1.73±0.03     | 2.79±0.01     |
| ViT (2x2 patch)          | 438±2.8       | 638±3.1       | 2.24±0.04     | 2.88±0.03     |
| SMoE after first layer   | 305±1.9       | 556±2.2       | 1.77±0.01     | 2.80±0.03     |
| Last layer SMoE          | 298±2.6       | 553±3.2       | 1.73±0.02     | 2.78±0.04     |
| 3x3 convs -> SMoE        | 278±2.0       | 530±1.8       | 1.69±0.01     | 2.65±0.01     |
|     + Gate prior         | **270±1.9**   | **525±2.0**   | **1.66±0.02** | **2.60±0.01** |
|     Rand fixed gate init | 328±3.7       | 572±4.1       | 1.89±0.08     | 2.96±0.05     |

A 2x wider (i.e., 256 filters per convolution) version of the Rasp & Thuerey ResNet offers little performance improvement. This is in line with their reported results, which showed decreasing improvements with wider networks. LRLCNs offer some improvements, but do not match SMoEs. Vision transformers do not perform well; we tried several different configurations and report the best.

Finally, following the suggestion of Reviewer vuZ5, we trained an SMoE model where the gates were fixed at the random initialization and not learned. This degraded performance significantly, as the selection of experts could not be adjusted and each expert saw fewer points in a sample. Hence, we can see that learning the gate is key.

---

### Meta-Review · Area_Chair_TH1r · 2022-08-26

**Recommendation:** Accept
**Confidence:** Less certain

**Metareview:**

This work introduces a novel layer in order to extract information better at a fine grained level. The basic idea is to introduce a spatial mixture of expert which will localized computations to a specific part of the image. Given the novelty the method, I recommend accepting this paper. However, please add the modifications proposed to reviewer vuZ5.

**Award:**

No

---

### Decision · Program_Chairs · 2022-09-14

Accept